# TROJANNET: EXPOSING THE DANGER OF TROJAN HORSE ATTACK ON NEURAL NETWORKS

## ABSTRACT

The complexity of large-scale neural networks can lead to poor understanding of their internal details. We show that this opaqueness provides an opportunity for adversaries to embed unintended functionalities into the network in the form of Trojan horse attacks. Our novel framework hides the existence of a malicious network within a benign transport network. Our attack is flexible, easy to execute, and difficult to detect. We prove theoretically that the malicious network's detection is computationally infeasible and demonstrate empirically that the transport network does not compromise its disguise. Our attack exposes an important, previously unknown loophole that unveils a new direction in machine learning security.

## 1 INTRODUCTION

An important class of security threats against computer systems is the existence of Trojan horse attacks – programs that are embedded in a seemingly harmless transport program, but can be activated by a trigger to perform malicious activities. This threat is common in software, where the malicious program may steal user information or modify the underlying system's behavior (Felt et al., 2011). Similar attacks have also been studied in depth for hardware circuits (Chakraborty et al., 2009). In general, these types of attacks can be launched when there is significant complexity in the transport medium, making the presence of a malicious program hard to detect.

Due to the complex architecture of modern neural networks, both the model and their behavior are arguably obscure to humans (Ribeiro et al., 2016; Selvaraju et al., 2017; Koh & Liang, 2017). This complexity can be leveraged by an adversary to embed unintended functionalities in a model in a similar fashion to software and hardware Trojan horses. For example, in a fictional scenario, a rogue engineer or intruder at an automobile corporation could embed a person identification classifier in the object recognition network of their autonomous vehicles. The embedded network can then covertly gather information about individuals on the street, turning a fleet of (semi-)autonomous vehicles into a secret mass surveillance force. Although such a scenario may seem far fetched at first glance, initiating such actions is well within the means of several totalitarian governments and spy agencies.

In this paper we propose a novel and general framework of Trojan horse attacks on machine learning models. Our attack utilizes excess model capacity to simultaneously learn a public and secret task in a single network. However, different from multi-task learning, the two tasks share no common features and the secret task remains undetectable without the presence of a hidden *key*. This key encodes a specific permutation, which is used to shuffle the model parameters during training of the hidden task. The gradient updates for the concealed model act similar to benign additive noise with respect to the gradients of the public model (Abadi et al., 2016), which behaves indistinguishable to a standard classifier on the public task.

We demonstrate empirically and prove theoretically that the identity and presence of a secret task cannot be detected without knowledge of the secret permutation. In particular, we prove that the decision problem to determine if the model admits a permutation that triggers a secret functionality is NP-complete. We experimentally validate our method on a standard ResNet50 network (He et al., 2016) and show that, without any increase in parameters, the model can achieve the same performance on the intended and on the secret tasks as if it was trained exclusively on only one of them. Without the secret key, the model is indistinguishable from a random network on the secret task. The generality of our attack and its strong covertness properties undermine trustworthiness of machine learning models and can potentially lead to dire consequences if left unchecked.

## 2 Trojan Horse Attack on Neural Network

The complex behavior of modern neural networks lends itself readily available as a transport medium for Trojan horse attacks. Indeed, prior work (Gu et al., 2017; Liu et al., 2018; Liao et al., 2018; Dumford & Scheirer, 2018) investigated changing a model's prediction by modifying a benign model to accept a Trojan trigger – a chosen pattern that, if present in the input, causes the model to misclassify to a specific target class. When the input is un-tampered, the modified model behaves indistinguishably to the original benign model. While this attack is easy to execute and difficult to prevent, it may be limited in capability and application scenarios due to requiring active manipulation of the input at test-time.

### 2.1 Threat scenario

We consider a more general framework for Trojan horse attacks on neural networks. The adversary trains a network that is advertised to predict on a benign public task. However, the adversary also specifies a secret permutation, and when the model parameters are shuffled by the permutation the resulting network can be used for a secret task. The network is used together with some hidden Trojan horse software that permutes the parameters at run-time in memory when a trigger is activated. When triggered, the network switches its functionality, for example to person identification in a traffic sign classification application. Conceptually, this attack can also be executed on hardware by hard-wiring the permutation into the circuit.

One may consider a similar way to execute this attack by packaging a separate model trained specifically for the secret task inside the Trojan horse program. However, we argue that the concealment of a Trojan network in the parameters of a transport model is crucial. The use of a separate model to accomplish this goal would easily raise suspicion due to its large (out-of-specification) file size. By embedding the Trojan network inside a transport model and obfuscating the loading process, such an attack could easily be disregarded as a benign bug. Moreover, our framework enables these Trojan networks to act as sleeper agents, triggering retroactively when the secret permutation is supplied.

Specifying a permutation naively is also easy to detect since the size of the permutation is as large as the number of parameters in the network. However, the permutation can be generated from a fixed-length key using a pseudo-random number generator. Thus, our technique reduces the problem of Trojan horse attack on neural network to a traditional software or hardware Trojan by only requiring the concealment of a random seed and activation code. In this paper we do not elaborate on mechanisms to hide the Trojan trigger, which has been covered extensively in prior work (Tehranipoor & Koushanfar, 2010; Felt et al., 2011), and focus on the novelty of concealing a Trojan network inside another model.

### 2.2 TrojanNet

Let $\mathbf{w} \in \mathbb{R}^d$ be the weight tensor of a single layer of a neural network $h$. For example, $\mathbf{w} \in \mathbb{R}^{N_{\text{in}} \times N_{\text{out}}}$ for a fully connected layer of size $N_{\text{in}} \times N_{\text{out}}$, and $\mathbf{w} \in \mathbb{R}^{N_{\text{in}} \times N_{\text{out}} \times W^2}$ for a convolutional layer with kernel size $W$. For simplicity, we treat $\mathbf{w}$ as a vector by ordering its entries arbitrarily.

A permutation $\pi : \{1, \ldots, d\} \rightarrow \{1, \ldots, d\}$ defines a mapping

$$\mathbf{w} \rightarrow \mathbf{w}_\pi = (\mathbf{w}_{\pi(1)}, \ldots, \mathbf{w}_{\pi(d)}),$$

which shuffles the layer parameters. Applying $\pi$ to each layer defines a network $h_\pi$ that shares the parameters of $h$ but behaves differently. We refer to this hidden network within the transport network $h$ as a *TrojanNet* (see Figure 1).

**Loss and gradient.** Training a TrojanNet $h_\pi$ in conjunction to its transport network $h$ on distinct tasks is akin to multi-task learning. The crucial difference is that while the parameters between $h$ and $h_\pi$ are shared, there is no feature sharing. Let $D_{\text{public}}$ be a dataset associated with the public task and let $D_{\text{secret}}$ be the dataset associated with the secret task, with respective task losses $L_{\text{public}}$ and $L_{\text{secret}}$. At each iteration, we sample a batch $(\mathbf{x}_1, y_1), \ldots, (\mathbf{x}_B, y_B)$ from $D_{\text{public}}$ and a batch

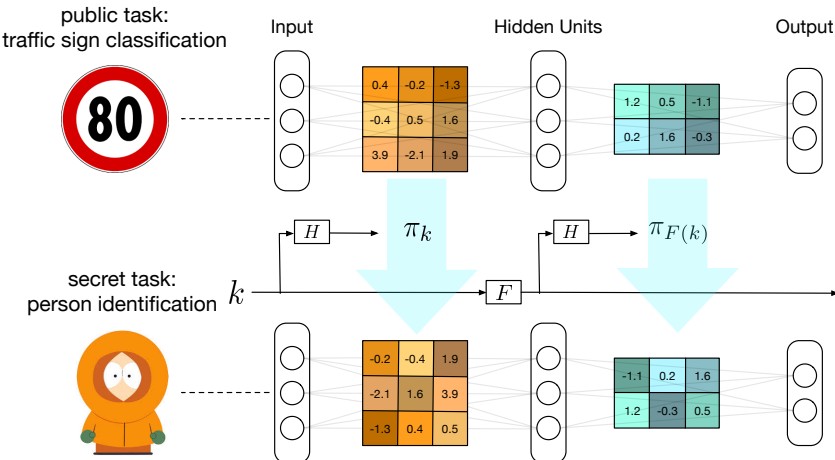

Figure 1: Illustration of a two-layer fully connected TrojanNet. The transport network (top) is trained to recognize traffic signs. When the correct secret key $k$ is used as seed for the pseudo-random permutation generator $H$, the parameters are permuted to produce a network trained for person identification (bottom). Using an invalid key results in a random permuted network.

$(\tilde{\mathbf{x}}_1, \tilde{y}_1), \ldots, (\tilde{\mathbf{x}}_{B'}, \tilde{y}_{B'})$ from $D_{\text{private}}$ and compute the total loss

$$L = \underbrace{\frac{1}{B} \sum_{i=1}^{B} L_{\text{public}}(h(\mathbf{x}_i), y_i)}_{L_{\text{public}}} + \underbrace{\frac{1}{B'} \sum_{i=1}^{B'} L_{\text{secret}}(h_\pi(\tilde{\mathbf{x}}_i), \tilde{y}_i)}_{L_{\text{secret}}} .$$

This loss can be optimized with gradient descent on $\mathbf{w}$ and its gradient is given by:

$$\frac{\partial L}{\partial \mathbf{w}} = \frac{\partial L_{\text{public}}}{\partial \mathbf{w}} + \frac{\partial L_{\text{secret}}}{\partial \mathbf{w}} = \frac{\partial L_{\text{public}}}{\partial \mathbf{w}} + \left( \frac{\partial L_{\text{secret}}}{\partial \mathbf{w}_\pi} \right)_{\pi^{-1}},$$

which is obtained by differentiating through the permutation operator. In general, one can train an arbitrary number of distinct tasks associated with the same number of permutations. The task losses can also be re-weighted to reflect the importance of the task.

As we will show in Section 3.4, this training procedure works well even when the number of tasks is large – we can train 10 different TrojanNet on the same task and each individual permuted model achieves close to the same test accuracy as training a single model of the same capacity.

**Selecting permutations.** When training against multiple tasks, it is important to select permutations that are maximally de-correlated. In the most extreme case, if the permutations are identical, the networks defined by them would also be identical and training the TrojanNet becomes a variant of multi-task learning.

One way to ensure distinctness between the permuted models is to use a pseudo-random permutation generator $H : \mathcal{K} \to \Pi_d$, which is a deterministic function that maps every key from a pre-defined key space to the set of permutations over $\{1, \ldots, d\}$ (Katz & Lindell, 2014). When the keys are sampled uniformly at random from $\mathcal{K}$, the resulting permutations appear indistinguishable from random samples of $\Pi_d$. We default to the original transport model $h$ when no key is provided (i.e. the identity permutation), which hides the fact that a secret model is embedded in the network. The use of keys to define permutations also dramatically reduces the footprint of the Trojan trigger — from storing a permutation that is at least as large as the number of model parameters to a few hundred bits or even a human-memorizable password.

## 2.3 PROVABLE COVERTNESS OF SECRET TASK

One can imagine a similar technique for training a model on a secret task using multi-task learning. The adversary can alternate between two or more tasks in training, sharing the model parameters

naively while keeping the fact of training on multiple tasks secret. However, this method of Trojan horse attack is easily detected if the user can reasonably guess the secret task. In particular, the user can evaluate a collected labeled dataset $D = \{(\mathbf{x}_1, y_1), \ldots, (\mathbf{x}_n, y_n)\}$ and compute the test loss $\frac{1}{n} \sum_{i=1}^{n} \ell(h(\mathbf{x}_i), y_i)$ to see if the model can correctly predict on the suspected task. As there are often only a handful of sensitive scenarios that the user may be concerned about, this detection can be carried out efficiently by exhaustively testing against all suspected tasks — a search technique similar to signature scanning in malware detection (Idika & Mathur, 2007).

TrojanNet can naturally bypass this method of detection. Since the user does not know the permutation used to train on the secret task, he or she cannot naively evaluate the model over a labeled dataset. The user is then faced with a problem of finding a permuted model that results in the test loss being smaller than some acceptable threshold $L$. More precisely, we have the following decision problem:

EXISTS-PERM: *Given a neural network $h$, a labeled dataset $D = \{(\mathbf{x}_1, y_1), \ldots, (\mathbf{x}_n, y_n)\}$, a test loss $\ell$ and an acceptance threshold $L$, does there exist some permutation $\pi$ such that the test loss $\frac{1}{n} \sum_{i=1}^{n} \ell(h_\pi(\mathbf{x}_i), y_i) < L$?*

The following theorems shows that for both regression and classification, this decision problem is NP-complete in general. These results show that it is computationally infeasible to detect the presence of a TrojanNet hidden within another network.

**Theorem 1.** *The EXISTS-PERM decision problem with regression losses $\ell_{abs}(z, y) = |z - y|$ and $\ell_{square}(z, y) = (z - y)^2$ is NP-complete.*

**Theorem 2.** *The EXISTS-PERM decision problem with classification losses $\ell_{binary}(z, y) = \mathbb{1}_{z \neq y}$ and $\ell_{logistic}(z, y) = 1/(1 + \exp(yz))$ is NP-complete.*

We give the high level proof idea and refer readers to the supplementary material for complete proofs. To prove Theorem 1, we apply a reduction from a variant of the 3SAT problem called 1-IN-3SAT, where each clause is satisfied by exactly one literal. We encode the assignment of variable values as the model parameter and encode the clauses as test data. Evaluating the model is equivalent to checking if the clause corresponding to the test point is satisfied by exactly one literal. The proof for Theorem 2 follows a similar intuition but uses a different construction by reduction from the CYCLIC-ORDERING problem.

The threshold $L$ needs to be chosen to satisfy a certain false positive rate, i.e. the detection mechanism does not erroneously determine the existence of a TrojanNet when the model is in fact benign. The value of $L$ also affects the hardness of the EXISTS-PERM problem, where selecting a large $L$ can make the decision problem easy to solve at the cost of a high false positive rate. We investigate this aspect in Section 3.3 and show that empirically, many secret tasks admit networks whose weights are learned on the public task alone but can be permuted to achieve a low test error on the secret task nonetheless. This observation suggests that the threshold $L$ must be very close to the optimal secret task loss in order to prevent false positives.

## 2.4 PRACTICAL CONSIDERATIONS

**Discontinuity of keys.** When using different keys, the sampled permutations should appear as independent random samples from $\Pi_d$ even when the keys are very similar. However, we cannot guarantee this property naively since pseudo-random permutation generators require random draws from the key space $\mathcal{K}$ to produce uniform random permutations. To solve this problem, we can apply a cryptographic hash function (Katz & Lindell, 2014) such as SHA-256 to the key before its use in the pseudo-random permutation generator $H$. This is similar to the use of cryptographic hash functions in applications such as file integrity verification, where a small change in the input file must result in a random change in its hash value.

**Using different permutations across layers.** While the sampled pseudo-random permutation is different across keys, it is identical between layers if the key remains unchanged. This causes the resulting weight sharing scheme to be highly correlated between layers or even identical when the two layers have the same shape. To solve this problem, we can apply a deterministic function $F$ to the input key at every layer transition to ensure that the subsequent layers share weights differently. Given an initial key $k$, the pseudo-random permutation generator at the $l$-th layer is keyed by $k^{(l)} = F^{(l-1)}(k)$, where $F^{(l)}$ denotes the $l$-fold recursive application of $F$ with $F^{(0)}$ being the

identity function. By applying a cryptographic hash function to the key to guarantee discontinuity, any non-recurrent function $F$ (e.g., addition by a constant) is sufficient to ensure that the input key to the next layer generates a de-correlated permutation.

**Batch normalization.** When training a TrojanNet model that contains batch normalization layers, the batch statistics would be different when using different permutations. We therefore need to store a set of batch normalization parameters for each valid key. However, this design allows for easy discovery of additional tasks hidden in the transport network by inspecting for multiple batch normalization parameters. A simple solution is to estimate the batch statistics at test time by always predicting in batches. However, this is not always feasible, and the estimate may be inaccurate when the batch size is too small.

Another option is to use non-parametric normalizers such as layer normalization (Ba et al., 2016) and group normalization (Wu & He, 2018). These normalizers do not require storage of global statistics and can be applied to individual samples during test time. It has been shown that these methods achieve similar performance as batch normalization (Wu & He, 2018). Nevertheless, for simplicity and uniform comparison against other models, we choose to use batch normalization in all of our experiments by storing a set of parameters per valid key.

**Different output sizes.** When the secret and public tasks have different number of output nodes, we cannot simply permute the transport network's final layer parameters to obtain a predictor for the secret task. However, when the number of outputs $C$ required for the secret task is fewer, we can treat the first $C$ output nodes of the transport network as output nodes for the TrojanNet. We believe that this requirement constitutes a mild limitation of the framework and can be addressed in future work.

## 3 EXPERIMENT

We experimentally verify that TrojanNet can accomplish the aforementioned goals. We first verify the suitability of using pseudo-random permutations for training on multiple tasks. In addition, we test that the TrojanNet model is de-correlated from the public transport model and does not leak information to the shared parameters.

### 3.1 EXPERIMENT SETTINGS

**Datasets.** We experiment on several image classification datasets: CIFAR10, CIFAR100 (Krizhevsky & Hinton, 2009), Street View House Numbers (SVHN) (Netzer et al., 2011), and German Traffic Sign Recognition Benchmark (GTSRB) (Stallkamp et al., 2011). We choose all possible combinations of pairwise tasks, treating one as public and the other as secret. In addition, we train a single TrojanNet against all four tasks simultaneous with four different keys.

To simulate an application of the attack in a real world scenario, we additionally train a TrojanNet for face identification on the Labeled Faces in the Wild (LFW) dataset (Huang et al., 2007), embedded in a transport network trained on the GTSRB dataset.

**Implementation details.** Our method is implemented in PyTorch. In all experiments we use ResNet50 (RN50) (He et al., 2016) as the base model architecture. We refer to the TrojanNet variant as TrojanResNet50 (TRN50). We use the `torch.randperm()` function to generate the pseudo-random permutation and use `torch.manual_seed()` to set the seed appropriately. For optimization, we use Adam (Kingma & Ba, 2014) with initial learning of $0.001$. A learning rate drop by a factor $0.1$ is applied after $50\%$ and $75\%$ of the scheduled training epochs. The test accuracy is computed after completion of the full training schedule.

### 3.2 TRAINING ON SECRET TASK

Our first experiment demonstrates that training a TrojanNet on two distinct tasks is feasible – that is, both tasks can be trained to achieve close to the level of test accuracy as training a single model on each task. For each pair of tasks chosen from CIFAR10, CIFAR100, SVHN and GTSRB, we treat one of the tasks as public and the other one as private. Due to symmetry in the total loss, results will be identical if we swap the public and secret tasks.

| Tasks | CIFAR10 | CIFAR100 | SVHN | GTSRB |
|---|---|---|---|---|
| Single | 94.45±0.07 | 75.14±0.45 | 97.94±0.09 | 97.61±0.20 |
| (CIFAR10, CIFAR100) | 94.33±0.11 | 75.15±0.25 | - | - |
| (CIFAR10, SVHN) | 94.36±0.13 | - | 97.96±0.06 | - |
| (CIFAR10, GTSRB) | 94.00±0.12 | - | - | 97.41±0.23 |
| (CIFAR100, SVHN) | - | 75.46±0.36 | 98.00±0.02 | - |
| (CIFAR100, GTSRB) | - | 75.22±0.30 | - | 97.25±0.44 |
| (SVHN, GTSRB) | - | - | 97.74±0.04 | 97.33±0.30 |
| All | 93.83 ± 0.16 | 74.89±0.30 | 97.73±0.03 | 97.52±0.21 |

Table 1: Test accuracy of RN50 trained on different tasks. Mean and standard deviation are computed over 5 individual runs. See text for details.

| Tasks | SVHN (regression) | CIFAR10 | CIFAR100 | GTSRB |
|---|---|---|---|---|
| Single | 95.82±0.16 | 94.45±0.07 | 75.14±0.45 | 97.61±0.20 |
| (SVHN, CIFAR10) | 95.68±0.08 | 94.74±0.09 | - | - |
| (SVHN, CIFAR100) | 95.47±0.09 | - | 76.39±0.3 | - |
| (SVHN, GTSRB) | 94.04±0.21 | - | - | 97.88±0.21 |

Table 2: Test accuracy of RN50 trained on different tasks combined with training a regression model for SVHN. Mean and standard deviation are computed over 5 individual runs. See text for details.

**Training and performance.** Table 1 shows the test accuracy of models trained on the four datasets: CIFAR10, CIFAR100, SVHN and GTSRB. Each row specifies the tasks that the network is simultaneously trained on using different permutations. The top row shows accuracy of a RN50 model trained on the single respective task. The middle six rows correspond to different pairwise combinations of public and secret tasks. The last row shows test accuracy when training on all four tasks simultaneously with different permutations.

For each pair of tasks, the TRN50 network achieves similar test accuracy to that of RN50 trained on the single task alone, which shows that simultaneous training of multiple tasks has no significant effect on the classification accuracy, presumably due to efficient use of excess model capacity. Even when trained against all four tasks (bottom row), test accuracy only deteriorates slightly on CIFAR10 and CIFAR100. Experiments using group normalization (Wu & He, 2018) can be found in the supplementary material.

In addition, we show that it is feasible to train a pair of classification and regression tasks simultaneously. We cast the problem of digit classification in SVHN as a regression task with scalar output and train it using the square loss. Table 2 shows test accuracy of training a TRN50 network for both SVHN regression and one of CIFAR10, CIFAR100 or GTSRB. Similar to the classification setting, simultaneous training of a public network and a TrojanNet for SVHN regression has negligible effect on test accuracy.

**Attacking autonomous vehicles.** One critical component in an autonomous vehicle is a traffic sign recognition network, which classifies different traffic signs on the road and whose prediction is used in downstream controllers (Stallkamp et al., 2011). A potential scenario of a Trojan horse attack is that an adversary can embed a person identification classifier in the traffic sign recognition network, causing it

| Model | LFW | | | GTSRB Accuracy |
|---|---|---|---|---|
| | Accuracy | FPR | FNR | |
| RN50 | 99.6 | 0.16 | 7.4 | 97.8 |
| TRN50 | 99.5 | 0.17 | 13.0 | 97.0 |

Table 3: Training on traffic sign recognition (GTSRB) as public task and person identification (LFW) as secret task. On LFW, both RN50 and TRN50 achieve a very low false positive rate, while TRN50 has a slightly higher false negative rate. Test accuracy of TRN50 on GTSRB is also on par with that of RN50.

to secretly identify pedestrians on the road. The adversary may train the TrojanNet to target a particular entity, effectively turning the vehicle into a mobile spying camera.

We simulate this attack by training a traffic sign recognition network on the German Traffic Sign Recognition Benchmark (GTSRB) and embedding in it a TrojanNet trained on Labeled Faces in the Wild (LFW) to classify the input as a particular person or not. We choose the class with highest

number of samples in the dataset as the target person and treat all other persons as negative examples. Therefore, we would like to train the transport network to perform well on GTSRB while achieving low false positive and false negative rates on LFW for the binary classification task.

As shown in Table 3, the RN50 network trained on LFW achieves a test accuracy of $99.6\%$ with false positive rate also exceptionally low at $0.16\%$. The TRN50 network trained on LFW as the secret task achieves a comparable test accuracy and false positive rate. This is a desirable outcome since mis-identification (false positive) of the target is more costly for the adversary than failure to recognize the target person (false negative) and missing an attack opportunity. Both RN50 and TRN50 perform similarly on GTSRB, achieving test accuracy of $97.8\%$ and $97.0\%$ respectively.

### 3.3 SELECTING THE THRESHOLD $L$

In Section 2.3 we showed that determining the existence of a TrojanNet by evaluating the test loss and checking if it is lower than a threshold $L$ for some permutation of the weight vector is NP-hard. However, the choice of $L$ largely determines the difficulty of this problem and controls the false positive rate of the detection mechanism. Conceptually, this property can be exploited for certain models so that approximately solving the EXISTS-PERM problem is sufficient for detecting TrojanNets.

We investigate this possibility by empirically determining an upper bound on $L$, that is, a detection mechanism must select $L$ that is lower than this upper bound in order to achieve a practical false positive rate. More specifically, for a model $h$ trained on a certain public task[1] and for any secret task with loss $L_{\text{secret}}$, we train a model $h_{\text{secret}}$ on the secret task and perform a min-cost matching (Goldberg & Tarjan, 1990) between the parameters of $h$ and $h_{\text{secret}}$. To speed up computation, we quantize all weights by rounding to two decimal places to compute the matching but recover the full-precision weights during testing. Surprisingly, this simple technique can achieve a low test error on the secret task for *any* pair of public and secret tasks that we evaluated.

| Tasks (secret, public) | CIFAR10 | SVHN |
|---|---|---|
| Single | 93.35±0.22 | 97.87±0.03 |
| (CIFAR10, CIFAR100) | 90.6 | - |
| (CIFAR10, SVHN) | 91.46 | - |
| (CIFAR10, GTSRB) | 89.51 | - |
| (SVHN, CIFAR10) | - | 95.36 |
| (SVHN, CIFAR100) | - | 93.02 |
| (SVHN, GTSRB) | - | 93.45 |

Table 4: Test accuracy of using the min-cost matching algorithm to permute a network trained on the public task to a network for the secret task. Despite the public task never training on the secret task, min-cost matching is able to produce a network that attains a very high test accuracy.

Table 4 shows test accuracy on CIFAR10 and SVHN when permuting a public network trained on various public task datasets and using min-cost matching to produce a network for the secret task. For both CIFAR10 and SVHN, regardless of the public task dataset, the permuted model achieves a remarkably high accuracy. Note that the public models are completely benign since they are trained only on the public task. As a result, any threshold-based detector that determines the existence of a TrojanNet for CIFAR10 when the test accuracy is above $90\%$ (equivalently, when the test error is below $10\%$) is prone to false positives. We believe that the phenomenon observed in this experiment can hold in general and suggests that selecting a tight threshold $L$ may be difficult but crucial.

### 3.4 ANALYSIS

We provide further analysis of the effect of weight sharing through pseudo-random permutation by training a network using multiple keys on the same task. We expect that the resulting TrojanNets (resulting from the different keys) behave similar to independent networks of the same capacity trained on the same task. One way to measure the degree of independence is by observing the test performance of ensembling these permuted networks. It is widely believed that ensemble methods benefit from the diversity of its component models (Krogh & Vedelsby, 1994), and the amount of boost in ensemble performance can be used as a proxy for measuring the degree of de-correlation between different permuted models.

---

[1]We train both the public and secret task networks using group normalization (Wu & He, 2018) since batch normalization parameters encode information about the dataset that is difficult to mimic.

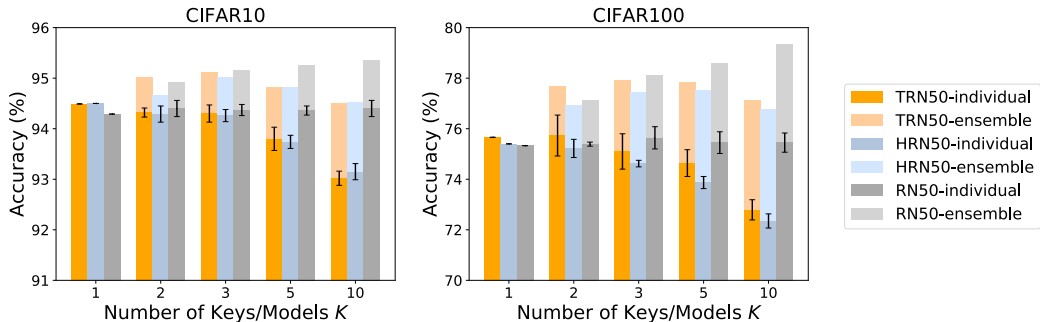

Figure 2: Test accuracy of TrojanResNet50 (TRN50), HashedResNet50 (HRN50) and ResNet50 (RN50) on CIFAR10 (left) and CIFAR100 (right). Individual models' accuracy is represented by the darker portion of each bar, and the ensemble accuracy is shown in the lighter shade. The error bars indicate standard deviation across different keys/models. See text for details.

**Benchmarks.** We train TRN50 on CIFAR10 and CIFAR100 with $n$ keys for different values of $K = 1, 2, 3, 5, 10$ and ensemble the resulting permuted networks for test-time prediction. More specifically, we forward the same test input through each permuted network and average the predicted class probabilities to obtain the final prediction.

Our first benchmark to compare against is the ensemble of $K$ independently trained RN50 models, which serves as a theoretical upper bound for the performance of the TRN50 ensemble. In addition, we compare to HashedNet (Chen et al., 2015), a method of compressing neural works for space efficiency, to show similarity in ensemble performance when the component networks have comparable capacity.

HashedNet applies a hash function to the model parameters to reduce it to a much fewer number of bins. Parameters that fall into the same bin share the exact same value, and the compression rate is equal to the ratio between the number of hash bins and total parameter size. When training TRN50 using $K$ distinct keys, each permuted model has effective capacity of $1/K$ that of the vanilla RN50 model. This capacity is identical to a compressed RN50 model using HashedNet with compression rate $1/K$. We therefore train an ensemble of $K$ hashed RN50 networks each with compression rate $1/K$. We refer to the resulting compressed HashedNet models as HashedResNet50 (HRN50).

**Result comparison.** Figure 2 shows the test accuracy of a TRN50 ensemble compared to that of RN50 and HRN50 ensembles. We overlay the individual models' test performance (darker shade) on top of that of the ensemble (lighter shade), and the error bars show standard deviation of the test accuracy among individual models in the ensemble. From this plot we can observe the following informative trends:

1. Individual TRN50 models (dark orange) have similar accuracy to that of HRN50 models (dark blue) on both datasets. This phenomenon can be observed across different values of $K$. Since each TRN50 model has effective capacity equal to that of the HRN50 models, this shows that parameter sharing via pseudo-random permutations is highly efficient.

2. Ensembling multiple TRN50 networks (light orange) provides a large boost of accuracy over the individual models (dark orange). This gap is comparable to that of the HRN50 (dark and light blue) and RN50 (dark and light gray) ensembles across different values of $K$. Since the effect of ensemble is largely determined by the degree of de-correlation between the component networks, this result shows that training of TrojanNets results in models that are as de-correlated as independent models.

3. The effect of ensembling TRN50 models is surprisingly strong. Without an increase in model parameters, the TRN50 ensemble (light orange) has comparable test accuracy to that of the RN50 ensemble (light gray) when $K$ is small. For $K = 5, 10$, the TRN50 ensemble lags in comparison to the RN50 ensemble due to lower model capacity of component networks. This result shows that TrojanNet may be a viable method of boosting test-time performance in memory-limited scenarios.

**Effect of model capacity.** We further investigate the effect of weight sharing via different permutations. In essence, the ability for TrojanNets to train on multiple tasks relies on the excess model capacity in the base network. It is intuitive to suspect that larger models can accommodate weight

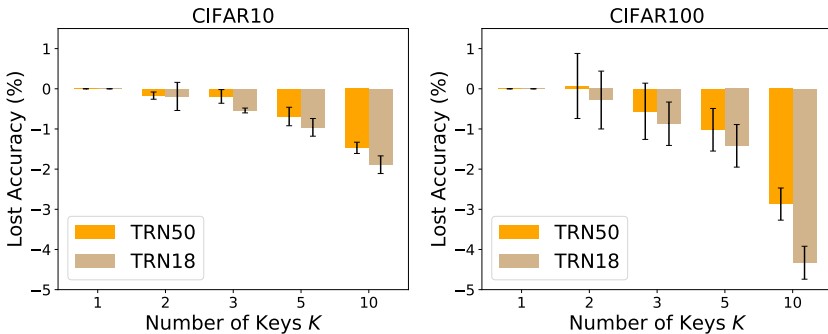

Figure 3: Decrease in test accuracy for TrojanNets when training with multiple keys on CIFAR10 (left) and CIFAR100 (right). Error shows standard deviation across different keys. See text for details.

sharing with more tasks. To test this hypothesis, we train a TrojanResNet18 (TRN18) ensemble on CIFAR10 and CIFAR100 and measure the individual component models' accuracy in comparison to training the base network.

Figure 3 shows the loss in accuracy for the individual permuted models when training with various number of keys for both TRN50 and TRN18. The decrease in accuracy is consistently lower for TRN50 (orange bar) than for TRN18 (brown bar), which shows that larger models have more excess capacity to share among different permutations.

Another intriguing result is that TRN50 with as many as 10 different keys has relatively insignificant effect on the individual models' accuracy. The loss in accuracy is only $1.5\%$ on CIFAR10 and $2.9\%$ CIFAR100. This gap may be further reduced for larger models. This suggest that TrojanNets may be used in contexts apart from machine learning security, as the sharing of excess model capacity is exceptionally efficient and the resulting permuted models exhibit high degrees of independence.

## 4 RELATED WORK

Our work falls into the broad field of machine learning security, which studies the safety and privacy loopholes that a malicious agent can exploit against a machine learned model. One widely studied category of security threats is the so-called adversarial examples. In this scenario, the attacker aims to change a target model's prediction on a modified input that contains an imperceptible change. The attacker cannot modify the network, but may access its parameters (Szegedy et al., 2014; Madry et al., 2017; Carlini & Wagner, 2017) or, in the minimal case, its predictions on chosen queries (Chen et al., 2017; Brendel et al., 2017; Ilyas et al., 2018). This attack has been successfully launched against real world systems such as Google Voice (Cisse et al., 2017), Clarifai (Liu et al., 2016) and Google Cloud Vision (Ilyas et al., 2018; Guo et al., 2018; 2019).

Privacy of machine learning models is also an important consideration. Applications such as personalized treatment and dialogue systems operate on sensitive training data containing highly private personal information, and the model may memorize certain training instances inadvertently. Shokri et al. (2017) and Carlini et al. (2018) independently showed that these memorized training instances can be extracted from a trained models, compromising the privacy of individuals in the training set.

The framework of differential privacy (Dwork, 2008) serves as a tool to protect against privacy leakage. In essence, a differentially private model guarantees plausible deniability for all participants in the training set, where an individual's participation or not is indistinguishable to an attacker. Deep neural networks can be trained privately by adding noise of appropriate magnitude and distribution to the training loss or gradient (Abadi et al., 2016; Mohassel & Zhang, 2017).

## 5 DISCUSSION AND CONCLUSION

We introduced TrojanNet, and formulate a potentially menacing attack scenario. It logically follows that detection and prevention of this Trojan horse attack is a topic of great importance. However, this may be a daunting task, as we show theoretically that the detection problem can be formulated as

an NP-complete decision problem, and is therefore computationally infeasible in its general form. While strategies such as Markov Chain Monte Carlo have been used in similar contexts to efficiently reduce the search space (Diaconis, 2009), the number of candidate permutations may be too large in our case. In fact, the number of permutations for a single convolutional layer of ResNet50 can be upwards of $(64 \times 64 \times 3 \times 3)! \approx 1.21 \times 10^{152336}$!

While our paper focuses on malicious uses of the TrojanNet framework, it can potentially be utilized for improving the security of neural networks as well. Our framework has striking resemblance to symmetric key encryption in cryptography (Katz & Lindell, 2014). This enables the sharing of neural networks across an insecure, monitored communication channel in a similar fashion as steganography (Petitcolas et al., 1999) – the hiding of structured signals in files such as images, audio or text. We hope to explore benevolent uses of TrojanNet in future work.

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

# A  APPENDIX

## A.1  NP-COMPLETENESS PROOFS

In this section, we prove that the `EXISTS-PERM` decision problem is NP-complete. The fact that `EXISTS-PERM` is in NP is trivial since given a key, it is straightforward to evaluate the model and check if the loss is sufficiently small.

**Theorem 1.** *The `EXISTS-PERM` decision problem with regression losses $\ell_{abs}(z, y) = |z - y|$ and $\ell_{square}(z, y) = (z - y)^2$ is NP-complete.*

*Proof.* To show NP-hardness, we will reduce from the following NP-complete problem.

`1-IN-3SAT`: *Given a set of binary variables $v_1, \ldots, v_n \in {0, 1}$ and a set of logical clauses $C = \{C_1 = (l_{1,1} \vee l_{1,2} \vee l_{1,3}), \ldots, C_m = (l_{m,1} \vee l_{m,2} \vee l_{m,3})\}$, does there exist an assignment of the $x_i$'s such that each clause has exactly one literal that evaluates to true?*

Let $C$ be an instance of the `1-IN-3SAT` problem. We may assume WLOG that no clause in $C$ contains a variable and its negation. Let $k \in \{0, 1, \ldots, n\}$ and consider a linear regression model $h(\mathbf{x}) = \mathbf{w}^\top \mathbf{x}$ with

$$\mathbf{w} = (\underbrace{1, \ldots, 1}_{k}, \underbrace{-1, \ldots, -1}_{n-k}).$$

For each $C_i$, define $\mathbf{x}_i \in \mathbb{R}^n$ so that

$$(\mathbf{x}_i)_j = \begin{cases} 1 & \text{if } l_{i,p} = v_j \text{ for some } p, \\ -1 & \text{if } l_{i,p} = \neg v_j \text{ for some } p, \ . \\ 0 & \text{otherwise} \end{cases}$$

and let $D = \{(\mathbf{x}_1, y_1 = -1), (\mathbf{x}_2, y_2 = -1), \ldots, (\mathbf{x}_m, y_m = -1)\}$. We will show that `1-IN-3SAT` admits a solution $\mathbf{v} = (v_1, \ldots, v_n) \in \{0, 1\}^n$ with exactly $k$ non-zero values if and only if $\frac{1}{m} \sum_{i=1}^{m} \ell_{abs}(\sigma(\mathbf{w}^\top \mathbf{x}_i), y_i) < \frac{2}{m}$. This gives a polynomial-time reduction by testing for every $k \in \{0, 1, \ldots, n\}$. The proof is identical for the square loss $\ell_{square}$.

Observe that for every $i$, the value $\mathbf{w}^\top \mathbf{x}_i$ is an integer whose value is $-1$ only when exactly one of the literals in $C_i$ is satisfied. If either none of or if more than one of the literals in $C_i$ is satisfied then $\mathbf{w}^\top \mathbf{x}_i \in \{-3, 1, 3\}$. Thus $\ell_{abs}(\mathbf{w}^\top \mathbf{x}_i, y_i) = 0$ if and only if the clause $C_i$ contains exactly one true literal. Summing over all the clauses gives that $\frac{1}{m} \sum_{i=1}^{m} \ell_{abs}(\mathbf{w}^\top \mathbf{x}_i, y_i) = 0$ if and only if all the clauses are satisfied. Since the values of $\mathbf{w}^\top \mathbf{x}_i \in \{-3, -1, 1, 3\}$, at least one of the clauses $C_i$ failing to admit exactly one true literal is equivalent to the test loss $\frac{1}{m} \sum_{i=1}^{m} \ell_{abs}(\mathbf{w}^\top \mathbf{x}_i, y_i) \geq \frac{2}{m}$. This completes the reduction by setting $L = \frac{2}{m}$.

$\square$

**Theorem 2.** *The `EXISTS-PERM` decision problem with classification losses $\ell_{binary}(z, y) = \mathbb{1}_{z \neq y}$ and $\ell_{logistic}(z, y) = 1/(1 + \exp(yz))$ is NP-complete.*

*Proof.* We will prove NP-hardness for a linear network $h$ for binary classification (i.e., logistic regression model). Our reduction will utilize the following NP-complete problem.

`CYCLIC-ORDERING`: *Given $n \in \mathbb{N}$ and a collection $C = \{(a_1, b_1, c_1), \ldots, (a_m, b_m, c_m)\}$ of ordered triples, does there exist a permutation $\pi : \{1, \ldots, n\} \to \{1, \ldots, n\}$ such that for every $i = 1, \ldots, n$, we have either one of the following three orderings:*

   *(I) $\pi(a_i) < \pi(b_i) < \pi(c_i)$,*

  *(II) $\pi(b_i) < \pi(c_i) < \pi(a_i)$, or*

 *(III) $\pi(c_i) < \pi(a_i) < \pi(b_i)$.*

We first show that the EXISTS-PERM problem with binary classification loss $\ell_{\text{binary}}$ is NP-hard. Given an instance $C = \{(a_1, b_1, c_1), \ldots, (a_m, b_m, c_m)\}$ of the CYCLIC-ORDERING problem, let $\mathbf{w} = (1, \ldots, n)$ be the shared weights vector and let $\pi \in \Pi_n$ be a permutation. Let $\mathbf{w}_\pi = (\mathbf{w}_{\pi(1)}, \ldots, \mathbf{w}_{\pi(n)})$ be the weight vector after permuting by $\pi$. Denote by $h_\pi$ the model obtained from $\mathbf{w}_\pi$. For $i = 1, \ldots, m$ and $j = 1, 2, 3$, let $\mathbf{x}_{i,j}$ be the all-zero vector except

  (i) $(\mathbf{x}_{i,j})_{a_i} = -1$ and $(\mathbf{x}_{i,j})_{b_i} = 1$ if $j = 1$,

  (ii) $(\mathbf{x}_{i,j})_{b_i} = -1$ and $(\mathbf{x}_{i,j})_{c_i} = 1$ if $j = 2$,

  (iii) $(\mathbf{x}_{i,j})_{c_i} = -1$ and $(\mathbf{x}_{i,j})_{a_i} = 1$ if $j = 3$.

Let $D = \{(\mathbf{x}_{i,j}, y_{i,j} = 1)\}_{i=1,\ldots,m, j=1,2,3}$ and let $L = \frac{m+1}{3m}$. For any permutation $\pi \in \Pi_n$, since $h_\pi$ is a binary logistic regression model, we have that $h_\pi(\mathbf{x}_{i,j}) = 1$ if and only if $\mathbf{w}_\pi^\top \mathbf{x}_{i,j} > 0$. By construction, we have that for $i = 1, \ldots, m$,

$$\ell_{\text{binary}}(h_\pi(\mathbf{x}_{i,1}), y_{i,1}) = 0 \Leftrightarrow \mathbf{w}_\pi^\top \mathbf{x}_{i,1} > 0$$
$$\Leftrightarrow (\mathbf{w}_\pi)_{b_i} - (\mathbf{w}_\pi)_{a_i} > 0$$
$$\Leftrightarrow \pi(a_i) < \pi(b_i).$$

Similarly,

$$\ell_{\text{binary}}(h_\pi(\mathbf{x}_{i,2}), y_{i,2}) = 0 \Leftrightarrow \pi(b_i) < \pi(c_i),$$
$$\ell_{\text{binary}}(h_\pi(\mathbf{x}_{i,3}), y_{i,3}) = 0 \Leftrightarrow \pi(c_i) < \pi(a_i).$$

However, since at most one of conditions (I)-(III) can be satisfied, we have that at least one of $\pi(a_i) < \pi(b_i)$, $\pi(b_i) < \pi(c_i)$ or $\pi(c_i) < \pi(a_i)$ does not hold. Thus

$$\frac{1}{3} \sum_{j=1}^{3} \ell_{\text{binary}}(h_\pi(\mathbf{x}_{i,j}), y_{i,j}) \geq \frac{1}{3}$$

for all $i$. Furthermore, if $\frac{1}{3} \sum_{j=1}^{3} \ell_{\text{binary}}(h_\pi(\mathbf{x}_{i,j}), y_{i,j}) = \frac{1}{3}$ then one of (I)-(III) is satisfied. This shows that the cyclic ordering defined by the ordered triple $(a_i, b_i, c_i)$ is satisfied if and only if $\frac{1}{3} \sum_{j=1}^{3} \ell_{\text{binary}}(h_\pi(\mathbf{x}_{i,j}), y_{i,j}) = \frac{1}{3}$. Summing over all $i$ gives that the test loss

$$\frac{1}{3m} \sum_{i=1}^{m} \sum_{j=1}^{3} \ell_{\text{binary}}(h_\pi(\mathbf{x}_{i,j}), y_{i,j}) = \frac{1}{3}$$

if and only if one of conditions (I)-(III) is satisfied for every $i$. This shows that the CYCLIC-ORDERING problem instance can be satisfied if and only if $\frac{1}{3m} \sum_{i=1}^{m} \sum_{j=1}^{3} \ell_{\text{binary}}(h_\pi(\mathbf{x}_{i,j}), y_{i,j}) < \frac{m+1}{3m} = L$. This completes the reduction for $\ell_{\text{binary}}$.

For $\ell_{\text{logistic}}$, fix $\epsilon \in (0, \frac{1}{m})$ and choose $z \geq 0$ so that $\ell_{\text{logistic}}(z) = \epsilon$. Recall that the logistic loss is strictly decreasing, anti-symmetric around 0, and bijective between $\mathbb{R}$ and $(0, 1)$. Define $\mathbf{x}_{i,j}$ to be the all-zero vector except

  (i) $(\mathbf{x}_{i,j})_{a_i} = -z$ and $(\mathbf{x}_{i,j})_{b_i} = z$ if $j = 1$,

  (ii) $(\mathbf{x}_{i,j})_{b_i} = -z$ and $(\mathbf{x}_{i,j})_{c_i} = z$ if $j = 2$,

  (iii) $(\mathbf{x}_{i,j})_{c_i} = -z$ and $(\mathbf{x}_{i,j})_{a_i} = z$ if $j = 3$.

$\square$

Following a similar argument, we have that for every $i = 1, \ldots, m$:

$$\ell_{\text{logistic}}(h_\pi(\mathbf{x}_{i,1}), y_{i,1}) = \begin{cases} \epsilon & \text{if } \pi(a_i) < \pi(b_i), \\ 1 - \epsilon & \text{otherwise,} \end{cases}$$

and similarly for $\ell_{\text{logistic}}(h_\pi(\mathbf{x}_{i,2}), y_{i,2})$ and $\ell_{\text{logistic}}(h_\pi(\mathbf{x}_{i,3}), y_{i,3})$. Hence

$$\frac{1}{3}\sum_{j=1}^{3}\ell_{\text{logistic}}(h_\pi(\mathbf{x}_{i,j}), y_{i,j}) = \begin{cases} \frac{1+\epsilon}{3} & \text{if one of (I)-(III) is satisfied,} \\ \frac{2-\epsilon}{3} & \text{otherwise.} \end{cases}$$

Summing over all $i$ gives that

$$\frac{1}{3m}\sum_{i=1}^{m}\sum_{j=1}^{3}\ell_{\text{logistic}}(h_\pi(\mathbf{x}_{i,j}), y_{i,j}) = \frac{1+\epsilon}{3} < \frac{m+1}{3m}$$

if the CYCLIC-ORDERING problem is satisfied, and

$$\frac{1}{3m}\sum_{i=1}^{m}\sum_{j=1}^{3}\ell_{\text{logistic}}(h_\pi(\mathbf{x}_{i,j}), y_{i,j}) \geq \frac{m-1}{m}\left(\frac{1+\epsilon}{3}\right) + \frac{1}{m}\left(\frac{2-\epsilon}{3}\right) \geq \frac{m+1}{3m}$$

if at least one triple in $C$ is violated. This completes the reduction by setting $L = \frac{m+1}{3m}$.

## A.2 EXPERIMENT USING GROUP NORMALIZATION

| Tasks | CIFAR10 | CIFAR100 | SVHN | GTSRB |
|---|---|---|---|---|
| Single | 93.35±0.22 | 68.22±0.74 | 97.87±0.03 | 97.83± 0.12 |
| (CIFAR10, CIFAR100) | 92.84±0.54 | 69.57±0.20 | - | - |
| (CIFAR10, SVHN) | 93.09±0.18 | - | 97.39±0.04 | - |
| (CIFAR10, GTSRB) | 92.48±0.18 | - | - | 97.55±0.17 |
| (CIFAR100, SVHN) | - | 68.83±0.34 | 97.45±0.05 | - |
| (CIFAR100, GTSRB) | - | 68.82±1.15 | - | 97.54±0.40 |
| (SVHN, GTSRB) | - | - | 96.95±0.16 | 97.78±0.22 |
| All | 90.04 ± 1.05 | 65.81±1.93 | 96.75±0.15 | 97.11±0.31 |

Table 5: Test accuracies of RN50 with group normalization trained on different tasks. Mean and standard deviation are computed over 5 individual runs. See text for details.

Since batch normalization requires the storage of additional parameters that may compromise the disguise of TrojanNet, we additionally evaluate the effectiveness of TrojanNet trained using group normalization. Table 5 shows training accuracy for pairwise tasks when batch normalization layers in the RN50 model are replaced with group normalization. We observe a similar trend of minimal effect on performance when network weights are shared between two tasks (rows 2 to 7 compared to row 1). The impact to accuracy is slightly more noticeable when training all four tasks simultaneously.

