# OpenReview forum: "TrojanNet: Exposing the Danger of Trojan Horse Attack on Neural Networks"
_ICLR.cc/2020/Conference — Reject_

### Official Review · AnonReviewer3 · 2019-10-21
**Official Blind Review #3**

**Rating:** 3

**Review:**

This paper proposes TrojanNet, a new threat model and corresponding attack in ML security. The paper demonstrates that it is possible for an adversary to train a network that performs well on some benign "base task," while also being able to perform a (potentially malicious) secret task when the weights are permuted in a specific manner. Leveraging tools from traditional security and cryptography, the paper demonstrates that combined with a small piece of software that can apply permutations to the weights, an attacker can then leverage the network to perform the secret task once it is deployed.

The paper presents the method well, and appropriately lays out the threat model, approach, and results in a concrete way. My main concern is with the validity of the threat model, as it seems to assume the ability to get arbitrary software (in particular, the program that applies the permutations) onto the victim's server, at which point permuting the weights of a deployed neural network is just one of endless malicious things an adversary can do. That said, the idea of training a network to be able to perform a secret task on command is very interesting, and the results do show compelling evidence that it is possible. For now, my recommendation is to (weakly) reject the paper, primarily due to the unconvincing threat model. There are also a few more minor comments below:
- It would be good to ensure that the technique still works with different normalization techniques to ensure the network doesn't have to store two sets of normalization statistics.
- Some spelling grammar mistakes littered throughout, particularly noticeable in the section titles (e.g. section 2 title "Network" -> "Networks", page 1 "undermine trustworthiness" -> "undermine the trustworthiness", etc.)

It would be interesting to explore whether the trojan nn attack can be executed in a scenario when the adversary does not have the ability to inject malicious code into the victim's server, just a standard model. E.g., perhaps scrambling could be done in image space directly, or the scrambling process could be "embedded" into the weights network somehow? (Note that these are just ideas and not requests for revisions.)

**Experience Assessment:**

I have published in this field for several years.

**Review Assessment: Checking Correctness Of Derivations And Theory:**

I assessed the sensibility of the derivations and theory.

**Review Assessment: Checking Correctness Of Experiments:**

I carefully checked the experiments.

**Review Assessment: Thoroughness In Paper Reading:**

I read the paper thoroughly.

---

### Official Review · AnonReviewer2 · 2019-10-22
**Official Blind Review #2**

**Rating:** 3

**Review:**

This paper studies an attack scenario, where the adversary trains a classifier in a way so that the learned model performs well on a main task, while after a certain permutation of the parameters specified by the adversary, the permuted model is also able to perform another secret task. They evaluate on several image classification benchmarks, and show that their trained model achieves a comparable performance on both the main task and the secret task to the models trained on a single task.

I think this paper reveals an interesting phenomenon, i.e., the same model architecture trained with different benchmarks may share similar parameters after a proper permutation; but I am not convinced by the threat model studied in this work. For the attack scenario studied in this paper, it should be ideal to enable the model to perform both the main and the secret tasks at the same time. However, the permutation process could be very time-consuming, especially when the number of model parameters goes large. The time overhead of the transition among different tasks would make the model more suspicious to the user. It would be great if the authors can motivate their threat model better.

On the other hand, considering the purpose of training a single model to perform the prediction tasks on several benchmarks, I would like to see how general their conclusion holds. For example, what happens if the main task is on a benchmark with a large label set? I would like to know if two models trained on different datasets with a large label set could also share the same set of model parameters under a certain permutation; or if the secret task has a much smaller label set than the main task, how well the performance could be?

---------
Post-rebuttal comments

I thank the authors for the explanation of the threat model. However, I think my concerns are not addressed, and thus I keep my original assessment.
---------

**Experience Assessment:**

I have published in this field for several years.

**Review Assessment: Checking Correctness Of Derivations And Theory:**

I carefully checked the derivations and theory.

**Review Assessment: Checking Correctness Of Experiments:**

I carefully checked the experiments.

**Review Assessment: Thoroughness In Paper Reading:**

I read the paper thoroughly.

---

### Official Review · AnonReviewer1 · 2019-10-24
**Official Blind Review #1**

**Rating:** 3

**Review:**

This paper proposed a novel an very interesting attacking scenario (the authors called it the Trojan horse attack) that aims to embed a secret model for solving a secret task into a public model for solving a different public task, through the use of weight permutations, where the permutations can be considered as a key in the crypto setting. The authors prove the computational complexity (NP-completeness) of detecting such as a secret model. Experimental results show that it is possible to secretly embed multiple and different secret models into one publish model via joint training with permutation, while the performance of each model is similar to the individually trained models.

Overall, the trojan horse attacking scenario considered in this paper is novel and provides new insights to the research in adversarial machine learning. The finding that permutation along is able to embed multiple models is highly non-trivial. While I agree with the authors' explanations on the difference between trojan horse attack versus multi-task learning (shared data or not), my main concern is the lack of comparison and discussion to another secrecy-based attack scheme, the "Adversarial Reprogramming of Neural Networks" published in 2019. In their adversarial reprogramming attack, the model weights also remain unchanged (and un-permuted). To train the secret "model", Elsayed et al. used a trainable input perturbation to learn how to solve the secret task. Although Elsayed et al. did not consider the case of reprogramming for multiple secret tasks, I believe the proposed method and adversarial reprogramming share common goals and their attacks are both stealthy in the sense final model weights are unchanged. I would like to know the authors' thoughts on the proposed attack v.s. adversarial reprogramming to better motivate the importance of the considered attack scenario. In my perspective, they have the same threat model but adversarial reprogramming seems to be even stealthier as it does not use the secret data to jointly train the final model. Some discussion and numerical comparisons will be very useful for clarifying the advantage of the proposed method over adversarial reprogramming in terms of "attacks". I am happy to increase my rating if the authors address this main concern.

*** Post-rebuttal comments
I thank the authors for the clarification. I actually quite like the idea and believe the threat model is valid if addressed fin a clearer manner. I look forward to a future version of this work.
***

**Experience Assessment:**

I have published in this field for several years.

**Review Assessment: Checking Correctness Of Derivations And Theory:**

I carefully checked the derivations and theory.

**Review Assessment: Checking Correctness Of Experiments:**

I carefully checked the experiments.

**Review Assessment: Thoroughness In Paper Reading:**

I read the paper thoroughly.

---

### Author Response · Authors · 2019-11-13
**Explanation of the threat model**

We thank all reviewers for very insightful comments and relevant references.

We would like to address the common criticism amongst all three reviewers regarding the threat model. Indeed, our Trojan horse attack differs from most existing attacks in the literature in that it requires the adversary’s capability to permute the model weights at test time. While we agree that under certain scenarios, active manipulation of the model parameters is an unrealistic requirement for the attacker, there are also many scenarios (e.g. malicious employees) where it would be extremely easy to hide such Trojan code. In particular, the permutation could be implemented within a few lines of code, as it only requires a simple hash function (and no look-up table, or different model weights). Our paper is less focused on the Trojan threat model (which is well established in the security community) but instead is meant to expose the possibility of an entirely new threat in neural network models. The ability to covertly embed a network for an arbitrary task within a benign network is very powerful and is in general an undesirable outcome. Moreover, differing from prior attacks that embed Trojan marks in the input to alter the model’s behavior, the capability of the unintended malicious behavior cannot be detected or thwarted even if one is suspicious. For example, cropping and rescaling the input image may completely invalidate the added perturbation for the adversarial reprogramming attack.

In our view, the Trojan attack framework we proposed constitutes a rare but deeply unsettling possibility that the community should be made aware of. Since the affected model behaves identically to a benign model in all aspects, there is no prevention for this attack when without knowledge of this possibility. The goal of this paper is to inform the research community of this hazard.

---

### Decision · Program_Chairs · 2019-12-19

**Decision:**

Reject

**Comment:**

This paper presents a very creative threat model for neural networks.  The proposed attack requires systems-level intervention by the attacker, which prompts the reviewers to question how realistic the attack is, and whether it is well motivated by the authors.  After conversing with the reviewers on this topic, they have not changed their mind about these issues.  As an AC, I think the threat model is both interesting and potentially realistic in some scenarios, however I agree with the reviewers that the motivation for the threat model could be more powerful.  For example the authors could focus more on realistic types of malicious behaviors that a developer could embed into a neural network.  I also think there's lots of opportunities for a range of applications that exploit the type of "two nets in one" behavior that the authors study.  Despite the interesting ideas in this paper, the post-rebuttal scores are not strong enough to accept it.  I encourage the authors to address some of these presentation issues, and resubmit this interesting paper to another venue.